# The Tryptophan Catabolite or Kynurenine Pathway in a Major Depressive Episode with Melancholia, Psychotic Features and Suicidal Behaviors: A Systematic Review and Meta-Analysis

**DOI:** 10.3390/cells11193112

**Published:** 2022-10-02

**Authors:** Abbas F. Almulla, Yanin Thipakorn, Asara Vasupanrajit, Chavit Tunvirachaisakul, Gregory Oxenkrug, Hussein K. Al-Hakeim, Michael Maes

**Affiliations:** 1Department of Psychiatry, Faculty of Medicine, Chulalongkorn University, Bangkok 10330, Thailand; 2Medical Laboratory Technology Department, College of Medical Technology, The Islamic University, Najaf 31001, Iraq; 3Cognitive Impairment and Dementia Research Unit, Faculty of Medicine, Chulalongkorn University, Bangkok 10330, Thailand; 4Department of Psychiatry, Tufts University School of Medicine and Tufts Medical Center, Boston, MA 02111, USA; 5Department of Chemistry, College of Science, University of Kufa, Kufa 54002, Iraq; 6Department of Psychiatry, Medical University of Plovdiv, 4002 Plovdiv, Bulgaria; 7School of Medicine, Barwon Health, IMPACT, The Institute for Mental and Physical Health and Clinical Translation, Deakin University, Geelong, VIC 3217, Australia

**Keywords:** melancholia, psychotic depression, affective disorders, neuro-immune, inflammation

## Abstract

Major depressive disorder (MDD) and bipolar disorder (BD) with melancholia and psychotic features and suicidal behaviors are accompanied by activated immune-inflammatory and oxidative pathways, which may stimulate indoleamine 2,3-dioxygenase (IDO), the first and rate-limiting enzyme of the tryptophan catabolite (TRYCAT) pathway resulting in increased tryptophan degradation and elevated tryptophan catabolites (TRYCTAs). The purpose of the current study is to systematically review and meta-analyze levels of TRP, its competing amino acids (CAAs) and TRYCATs in patients with severe affective disorders. Methods: PubMed, Google Scholar and SciFinder were searched in the present study and we recruited 35 studies to examine 4647 participants including 2332 unipolar (MDD) and bipolar (BD) depressed patients and 2315 healthy controls. Severe patients showed significant lower (*p* < 0.0001) TRP (standardized mean difference, SMD = −0.517, 95% confidence interval, CI: −0.735; −0.299) and TRP/CAAs (SMD = −0.617, CI: −0.957; −0.277) levels with moderate effect sizes, while no significant difference in CAAs were found. Kynurenine (KYN) levels were unaltered in severe MDD/BD phenotypes, while the KYN/TRP ratio showed a significant increase only in patients with psychotic features (SMD = 0.224, CI: 0.012; 0.436). Quinolinic acid (QA) was significantly increased (SMD = 0.358, CI: 0.015; 0.701) and kynurenic acid (KA) significantly decreased (SMD = −0.260, CI: −0.487; −0.034) in severe MDD/BD. Patients with affective disorders with melancholic and psychotic features and suicidal behaviors showed normal IDO enzyme activity but a lowered availability of plasma/serum TRP to the brain, which is probably due to other processes such as low albumin levels.

## 1. Introduction

Major depression disorder (MDD) and bipolar disorder (BD) may involve severe phenotypes including melancholia and psychotic depression [1]. Delusions and hallucinations are the main characteristics of the psychotic features in MDD, while melancholic MDD is characterized by severe depressed mood, anhedonia, hypoesthesia, lack of reactivity, early morning awakening, diurnal variation, and anorexia resulting in weight loss, and psychomotor retardation. These subtypes of MDD are strongly associated with suicidal behaviors in MDD patients [2]. Suicide is one of the major causes of death worldwide, one of each 100 deaths is due to suicide, and globally 800,000 individuals die per year [3,4]. The presence of psychiatric illness, particularly MDD and BD, is the major leading cause of suicide, particularly MDD (around 90% of the victims) [5,6].

Extensive evidence is now available indicating the involvement of activated immune-inflammatory and oxidative and nitrosative stress pathways (IO&NS) in the pathophysiology of mental disorders including MDD, BD and schizophrenia [7,8,9,10,11]. Moreover, MDD and BD are accompanied by activation of the immune-inflammatory response system (IRS) reflected by alterations of acute phase proteins (APPs) (e.g. C-reactive protein and albumin) and activation of cell-mediated immunity as shown by increased interleukin (IL)-6, tumor necrosis factor (TNF)-α, IL-1β, IL-2, interferon (IFN)-γ, soluble IL-2 receptor (sIL-2R), sCD8, high levels of activated T cells such as CD25+ and HLA-DR+) [11]. IRS is usually counterbalanced by the compensatory immune-regulatory system (CIRS), which increases T-regulatory cytokines such as IL-10 and transforming growth factor (TGF)-β [7,12,13,14]. A large body of studies indicates that MDD and BD are associated with elevated peripheral levels of lipopolysaccharide (LPS), which augment inflammation and cell-mediated immunity (CMI) [15,16].

In comparison with simple MDD, MDD with melancholic and psychotic features and suicidal behaviors are reported to be associated with increased pro-inflammatory markers, namely APPs (e.g., haptoglobin), upregulated T cell markers, besides failure to suppress the production of IL-1β and sIL-2R by administration of dexamethasone [2,17,18,19,20]. Hyperactive immune-inflammatory pathways may induce O&NS pathways which are accompanied by elevated reactive oxygen and nitrogen species (ROS, RNS), especially in case of lowered total antioxidant capacity levels [7]. Additionally, increased levels of myeloperoxidase (MPO), a key enzyme in the innate immune response, has been frequently reported in depression [21,22]. The latter may increase reactive chlorine species (RCS, e.g. hydrochlorous acid) resulting in chlorinative stress followed by high levels of advanced oxidation protein products (AOPP) [23]. Moreover, high levels of oxidative mediators impact the integrity of lipids, proteins, DNA, and mitochondria [7]. Stimulated IRS and O&NS pathways explain, in part, key characteristics of affective disorders, namely the frequency of episodes (disease’s staging), the severity of illness, and suicidal behaviors, including suicidal ideation and attempts [24,25,26]. Furthermore, the neurotoxic properties of ROS, RNS, and M1 macrophage and T helper (Th)-1 cytokines generate neuro-affective toxicity, which may explain the staging and phenome of MDD and BD [24,25].

High levels of IFN-γ, IL-1β, LPS, along with ROS and RNS are implicated in induction the rate-limiting enzyme of the tryptophan catabolite (TRYCAT) pathway, namely indole 2,3-dioxygenase (IDO) enzyme [27,28,29,30]. The TRYCAT pathway is the major catabolic pathway of tryptophan (TRP) which when overactive may deplete TRP thereby producing neuroactive metabolites, including kynurenine (KYN), kynurenic acid (KA), 3-hydroxykynurenine (3HK), anthranilic acid (AA), 3-hydroxyanthranilic acid (3-HA), xanthurenic acid (XA), quinolinic acid (QA), and picolinic acid (PA). The latter TRYCATs show neuroprotective as well as neurotoxic effects as shown in Figure 1 [30,31]. Besides, the depletion of central and peripheral TRP levels (a precursor of serotonin) may lower serotonin levels in the CNS which have been reported in impulsive suicidal patients [30,31,32,33]. Some TRYCATs cause neuro-oxidative toxicity with oxidative cell damage and lipid peroxidation, such as 3HA, 3HK and QA [28,30,34,35,36]. Additionally, a substantial amount of hydrogen peroxide and superoxide anions are produced by 3HA and 3HK [37].

Furthermore, frequent agonistic effects of QA on the hippocampal N-methyl-D-Aspartate (NMDA) receptors may induce atrophy and apoptosis of the hippocampus [31]. Elevated XA levels may cause neurotoxicity by overactivation of the cationic channels that lead to intracellular hypercalcemia and, hence, accelerate damage to neural circuits in the brain along with mitochondrial dysfunction and apoptosis. These processes may substantially damage the neurons, produce poor glutamate transmission and restrict presynaptic transmission via triggering NMDA receptors [30,38]. PA promotes immune-inflammatory response and reduces neuroprotection via reducing AA and KA levels which additionally have antidepressant roles [38,39]. In contrast, KYN may exert depressogenic and anxiogenic effects [31].

We recently found, in a large-scale meta-analysis, that affective disorders, including mild to moderate severe MDD and BD, are associated with a reduction in TRP levels but without any signs of an overactivated TRYCAT pathway [40]. However, in MDD patients with suicidal, melancholic, and psychotic features, central and peripheral reductions in TRP levels, activation of the TRYCAT pathway, and increased TRYCATs levels were frequently reported in previous studies, which may reveal upregulation of the IDO enzyme is the leading cause for TRP depletion and increased TRYCATs levels [41,42,43]. Nonetheless, the TRP, competing amino acids (CAAs) and TRYCATs levels were not systematically reviewed in MDD/BD patients with the most severe phenotypes. Thus, in the purpose of the current study is to systematically review and meta-analyze TRP, CAAs, and the activity of TRYCAT pathway as reflected by KYN/TRP ratio (IDO enzyme index), KA/KYN (kynurenine Aminotransferase, KAT enzyme index), neurotoxicity indices and solitary levels of TRYCATs in MDD/BD patients with melancholic or psychotic features and suicidal behaviors.

## 2. Material and Methods

In the present meta-analysis, we investigated TRP, CAAs, the TRP/CAAs ratio, KYN, KA, AA and QA levels, and KYN/TRP and KA/KYN ratios as indicators for IDO and KAT enzyme activities, respectively. Additionally, we also computed a neurotoxic composite, namely (KYN+3HK+3HA+QA+XA+PA). These biomarkers were examined in serum and plasma (peripherally), cerebrospinal fluid (CSF) and brain tissue (centrally) of patients with severe affective disorders who show features of melancholia, psychosis, or suicidal behavior versus healthy controls. This study followed the Preferred Reporting Items for Systematic Reviews and Meta-analyses (PRISMA) 2020 criteria [44], the Cochrane Handbook for Systematic Reviews and Interventions guidelines [45], and the Meta-Analyses of Observational Studies in Epidemiology (MOOSE) guidelines.

## 3. Search Strategy

Appendix A displays the keywords and MESH terms utilized to search the electronic databases, including PubMed/MEDLINE, Google Scholar, and SciFinder, for publications concerning TRP and TRYCATs in melancholia, psychotic features, and suicides of affective disorders. Moreover, to ensure the comprehensiveness of our search, we reviewed the reference lists of all eligible papers and prior meta-analyses. The current data collection processes extended from 10 January to 31 March 2022.

## 4. Eligibility Criteria

Articles in English that were published in peer-reviewed journals were included in our meta-analysis. Nevertheless, manuscripts in different other languages, namely Thai, French, Spanish, German, Italian, and Arabic, along with grey literature, were also selected. Other inclusion criteria were (a) observational case-control and cohort studies that employed serum, plasma, CSF and brain tissues to evaluate TRP, CAAs, and/or TRYCATs, (b) Diagnostic and Statistical Manual of Mental Disorders (DSM) or the International Classification of Diseases (ICD) must have been utilized to diagnose MDD and BD with either melancholia, psychotic feature and/or suicidal behavior, and (c) the number of patients with melancholia, psychotic features, or suicidal behavior should be reported. Exclusion criteria were: (a) genetic, animal-based and translational studies, (b) research that lacked a control group, (c) studies that utilized samples such as saliva, hair, whole blood, or platelet-rich plasma, and (d) duplicate articles along with systematic review and meta-analyses. We emailed the authors when they did not report the mean and standard deviation (SD) or standard error (SE) of the measured biomarkers. We employed the Wan et al [46] approach to estimate the mean (SD) and the median (interquartile range or range) when we did not receive a response from authors. If graphical data were provided, the mean and SD values were extracted using the Web Plot Digitizer (https://automeris.io/WebPlotDigitizer/).

## 5. Primary and Secondary Outcomes

As the primary outcomes, we examined TRP, TRP/CAAs, CAAs, KYN/TRP ratios and KYN levels in patients with affective disorders with melancholic or psychotic features, and/or suicidal behaviors versus healthy control. Secondary outcomes involved the KA/KYN ratio (KAT enzyme) and the neurotoxic composite score (KYN + 3HK + 3HA + QA + XA + PA) along with solitary KA, AA and QA levels.

### 5.1. Screening and Data Extraction

Using the above inclusion criteria, the first two authors (AA and YT) performed the elementary search processes. We checked the titles and abstracts of relevant manuscripts to evaluate including them in the current meta-analysis. Once the articles passed this checking step, we downloaded the full-text articles. The first author (AA) made a Microsoft Excel file to accommodate the extracted data, mainly mean (SD) and sample size of the assessed biomarkers. Furthermore, we also recorded the medium in which the analytes were determined (serum, plasma, CSF, brain tissues), type of affective disorder and whether they showed melancholic or psychotic features, and/or suicidal behaviors, authors’ names, publication dates, location of study, and study design, as well as sex, age of the participants, psychiatric ratings scales, and the psychiatric and physical comorbidities of patients and controls. The second and third authors (YT and AV) performed double-checks for the Excel file, and they consulted the last author (MM) is case of disagreements. The last author MM, adjusted the immunological confounder scales for TRP and TRYCATs studies to evaluate the methodological quality of the included articles [47]. Appendix A shows these two rating scales, namely quality and redpoint scales which were used to examine the quality of immune-based articles on schizophrenia [48], Alzheimer’s disease [49], Coronavirus disease 2019 [50] and affective disorders [40]. Sample size, confounder control, and the time of sampling were the main items of the quality scale, which ranged from 0 to 10, and the best quality is achieved when the score is close to 10. The redpoint scale examines the quality level of the study design in terms of major confounders, including biological and analytical bias, which can be detected by higher redpoint scale scores (ranging from 0 to 26).

### 5.2. Data Analysis

Appendix A shows the PRISMA criteria which were employed in the present meta-analysis that used the CMA program V3 to analyze all of the data. The criterion to conduct a meta-analysis was that the biomarker levels should be available in at least three studies. We assumed dependency while computing the mean values of the outcomes to compare the neurotoxicity index and the KYN/TRP (an index for IDO enzyme activity) and KA/KYN (an index for KAT enzyme activity) ratios in depressed patients versus healthy controls [48,49]. We evaluate the following ratios by specifying the effect size direction (a) KYN/TRP with KYN in positive direction and TRP in negative direction, (b) KA/KYN with KA in positive and KYN in negative direction; (c) TRP/CAAs ratio: TRP in positive and CAAs in negative direction. We employed the random-effects model with restricted maximum likelihood to extract the effect size and report the standardized mean difference (SMD) with 95 percent confidence intervals (95% CI) as an indicator for the effect size with a two-tailed *p*-value less than 0.05 to describe the statistical significance. According to the values of SMD, namely 0.2, 0.5, and 0.8, the effect sizes were considered small, medium, and large, respectively [50]. We used tau squared statistics to delineate heterogeneity in the data but also assessed the Q and I^2^ metrics [11,48,49]. We also performed meta-regression analyses to detect the sources of heterogeneity. Subgroup analysis was utilized to find the variations in TRP and TRYCATs among patients with melancholia, psychotic features, and suicidal behavior and central nervous system (CNS, brain tissues + CSF), serum and plasma, while selecting each of the latter groups as a unit of analysis. Since we did not find any significant difference between CSF and brain tissues, we combined the results from CSF and brain tissues into one group, called CNS. The effect sizes obtained from melancholic, psychotic, and suicidal patients were pooled in the absence of any significant difference between the above groups. Nevertheless, if there were significant intergroup differences, we report the effect sizes separately in the various subgroups. The strength of the effect sizes was examined by carrying out sensitivity analysis utilizing a leave-one-out approach. The fail-safe N technique along with one-tailed *p*-values for Kendall tau with continuity correction and Egger’s regression intercept were computed to investigate possible publication bias. The adjusted effect sizes were computed after imputing missing studies by the trim-and-fill method when the Egger’s test showed significant asymmetry. Funnel plots are generated and show study precision plotted versus SMD (with both observed and imputed values).

## 6. Results

### 6.1. Search Results

The number of included and excluded studies and the final search outcomes are displayed in the PRISMA flow chart, Figure 2. We employed MESH terms and keywords (all shown in Appendix A) to perform the initial search process, including inspection of 10861 articles. Based on our exclusion criteria 35 studies were selected as eligible in our systematic review and meta-analysis [41,42,43,51,52,53,54,55,56,57,58,59,60,61,62,63,64,65,66,67,68,69,70,71,72,73,74,75,76,77,78,79,80,81].

One of the included studies reported two separate cohorts of patients, namely those with melancholia and psychotic features. Therefore, we entered the study two times and pooled the overall effect sizes from 36 studies in the present systematic review and meta-analysis (8 CNS, 21 plasma, and 7 serum). We included 11 studies on depression with melancholia (9 plasma and 2 serum), 8 studies of depression with psychotic features (1 CNS, 5 plasma and 2 serum) and 17 studies with suicidal behavior (7 CNS, 7 plasma and 3 serum). The current meta-analysis included 4647 participants distributed as 2332 patients and 2315 healthy controls. The mean ages of the individuals in the studies extended from 30 to 59 years.

Appendix A shows that the USA, Belgium, and Sweden contributed most to the total number of studies (8, 6 and 5 studies), respectively. Norway, the United Kingdom, Germany, Netherlands, and South Korea contributed with 2 studies and Italy, Spain, Ireland, China and Tunisia contributed each 1 study. High-performance liquid chromatography (HPLC) has been used in 14 studies and was, therefore, the most common technique employed to assess TRP and TRYCATs (see Appendix A). This table also shows the quality (median = 5.62, min = 2.75, max = 7.75) and redpoint (median = 13.75, min = 9.5, max = 18.5) scores.

### 6.2. Primary Outcome Variables

#### 6.2.1. TRP, CAAs Levels and the TRP/CAAs Ratio

Table 1 shows that the effect size of the TRP level was pooled from 29 studies. The CI was completely on the negative side of zero in 13 studies, whereas only 2 studies showed that the CIs were on the positive side of zero. Furthermore, 14 studies intersected with zero with a negative SMD in 12 and a positive SMD in 2 studies. TRP levels were significantly decreased with a moderate effect size (SMD = −0.517) in patients compared to healthy controls. Figure 3 shows the forest plot of the TRP results. Publication bias analysis revealed 5 missing studies to the right side of the funnel plot and imputing these studies resulted in a lowered effect size although it remained significant.

CAAs results were obtained from 5 studies which involved only MDD with melancholia. Table 1 shows that the CIs of 2 studies fell entirely on the negative side of zero and 3 studies intersected with zero, 2 with negative and 1 with positive SMD values. Appendix A shows that CAAs were not significantly different between patients and controls.

The effect size of TRP/CAAs ratio was extracted from 7 studies performed in MDD with melancholia features. The 95% CI was entirely on the negative size of zero in 2 studies and 5 studies showed an overlap with zero. Melancholic MDD patients showed a significant reduction in TRP/CAAs with moderate effect size (SMD = −0.617). There were 2 missing studies on the left side of the funnel plot and after adjusting the effect size for these missing studies the SMD value was −0.748.

#### 6.2.2. The KYN/TRP Ratio and KYN Levels

Table 1 and Figure 4 revealed no significant differences in the KYN/TRP ratio between patients and controls. Due to the high heterogeneity, we performed group analysis which showed significant differences between melancholic, suicidal and psychotic patients and a significant increase with a small effect size was established in psychotic depression. Z Kendall’s and Egger’s test showed no signs of publication bias.

Table 1 and Table 2 and Appendix A show no significant difference in KYN levels between patients and controls. Group analysis displayed a significant difference (*p* = 0.041) between CNS, serum and plasma KYN levels which were significantly decreased in plasma. Table 3 revealed 3 missing studies on the right side of funnel plot and imputing these missing data yielded a non-significant effect size.

### 6.3. Secondary Outcome Variables

#### 6.3.1. Neurotoxicity Composite (KYN + 3HK + 3HA + XA + QA + PA) and KA/KYN Ratio

Table 1 and Appendix A show no significant differences in this neurotoxicity composite between patients and controls. We obtained the effect size of KA/KYN ratio from 14 studies and Table 2 and Appendix A indicate no overall difference between patients and controls. However, group analysis showed a significant difference between melancholic, psychotic, and suicidal patients, while only the latter displayed a significant decrease with a small effect size in the KA/KYN ratio compared to controls. Table 3 shows that there were two missing studies on the left side of the funnel plot of the KA/KYN ratio in suicidal patients and the adjusted estimate value was more decreased after imputing these missing studies.

#### 6.3.2. Solitary Levels of KA, AA and QA

Table 2 and Appendix A show that KA levels in patients were significantly lower with a small effect size in patients as compared with controls. Table 1 and Appendix A show that the effect size of AA was obtained from three studies and that AA is significantly decreased with small effect size in patients. Table 3 shows one missing study to the right side of the funnel plot and after imputing this missing study showed that the results were no longer significant. 

Table 1 and Table 2 and Appendix A show that the QA levels were significantly increased with small effect size in patients versus controls. Publication bias analysis showed that there were three missing studies on the right of the funnel plot and after imputing these studies, the effect size increased to 0.646.

#### 6.3.3. Meta-Regression Analyses

In order to examine factors which could explain heterogeneity, we carried out meta-regression analyses (Appendix A). Plasma was the most important confounder increasing heterogeneity with significant effects on TRP, KYN, KA, QA and (KYN + 3HK + 3HA + XA + QA + PA). Moreover, male/female gender, absence of electroconvulsive therapy (ECT), medications and latitude also impact heterogeneity.

## 7. Discussion

### 7.1. Availability of TRP to the Brain

The first major finding in the current study is that TRP levels and the TRP/CAAs ratio are significantly decreased in patients with MDD/BD with melancholia and psychotic features, and suicidal behaviors as compared with healthy controls and that there was no significant difference between the latter phenotypes in TRP and TRP/CAAs levels. The current results align with findings of a recent study conducted on MDD and BD patients with mild to moderate forms of depression [40]. Moreover, previous studies revealed a significant TRP reduction in affective disorders with melancholic, psychotic and suicidal features [55,61,62,66]. We also found that melancholic MDD patients have normal levels of CAAs and a decreased TRP/CAAs ratio, which is therefore solely determined by diminished TRP levels. These findings are consistent with our recent findings in mild to moderate MDD/BD [40]. We could not examine the CAAs levels and TRP/CAAs ratio in MDD/BD patients with psychotic features and/or suicidal behaviors because there were not sufficient studies. Thus, the current meta-analysis confirms previous studies about MDD patients with melancholia [63,64,65,66].

It is important to measure the TRP/CAAs ratio because both peripheral TRP (whether total or free) and CAAs (tyrosine, valine, phenylalanine, leucine and isoleucine) levels determine at least in part brain TRP concentration [82,83]. Indeed, specific receptors in the blood-brain barrier (BBB), named the large amino acid transporter 1 (LAT 1), are responsible for delivering TRP to the brain from the peripheral blood and the above CAAs compete with TRP to the cross BBB [48]. In this respect, Moller [84] reported that a decreased TRP/CAAs ratio predicted a successful response to the selective serotonin reuptake inhibitors (SSRIs).

### 7.2. KYN Levels and IDO Enzyme

The second major finding of the present study is that patients with severe affective disorders showed unchanged KYN levels compared with healthy controls. However, these results showed high heterogeneity, and, hence, we performed group analysis which revealed that only plasma KYN level was significantly decreased which is consistent with previous results in mild to moderate MDD/ BD [40,48]. We also found that the KYN/TRP ratio was unaltered in all patients combined although patients with psychotic features showed a trend towards significant increases in the KYN/TRP ratio. Nonetheless, since KYN is not significantly increased, such changes do not result from IDO activation.

While some prior studies are in line with our findings and showed neither significant changes in KYN/TRP ratio nor KYN level in suicidal patients [76,80], Messaoud et al. reported a significant plasma elevation of KYN level and KYN/TRP ratio in suicidal MDD patients [85]. However, another study showed a high serum KYN/TRP ratio without changes in KYN level [81]. The KYN/TRP ratio was significantly increased in MDD patients with psychotic features, while the KYN level decreased significantly [75]. Other studies that included melancholic patients reported no significant changes in plasma KYN levels and the KYN/TRP ratio [60,67].

The current meta-analysis indicates that there is no overactivation of the TRYCAT pathway in severe forms of affective disorders, which are, nevertheless, accompanied by a mild chronic immune-inflammatory response. Similar findings were recently reported in Alzheimer’s disease, another disorder accompanied by a mild chronic inflammatory process [40,49]. In contrast, the TRYCAT pathway is more pronounced in conditions characterized by severe acute inflammatory conditions such as COVID-19 infection [86] and treatment with IFN-α [87]. In those conditions, cytokine induced TRP depletion (also known as TRP starvation) is a key element in the innate immune response that impedes intruding pathogens and results in anti-inflammatory effects [30,31].

Nevertheless, several factors influence the activity of the IDO enzyme and probably lead to inhibition of the TRYCAT pathway. First, lowered TRP levels may drive the self-regulation of the IDO enzyme which is accompanied by an inactive ferric IDO form and autoxidation [88,89]. Second, elevated nitric oxide levels are reported in severe MDD and BD and may inhibit the IDO enzyme [90,91,92]. Third, cellular IDO is probably inhibited at the post-translational level by high hydrogen peroxides concentrations [93] which is another hallmark of depression [94]. Moreover, other substantial factors which influence the TRYCAT pathway are deficiencies of riboflavin (vitamin B2), a coenzyme of kynurenine 3-monooxygenase (KMO), and pyridoxal 5′-phosphate (PLP, vitamin B6) which is the coenzyme for KAT and kynureninase (KYNU) [95]. Both vitamins are repeatedly reported to be decreased in depression [96,97].

Since IDO is not activated in severe depression, other factors should explain the lower TRP availability to the brain. First, depression is accompanied by lowered levels of albumin, a negative APP, whilst a large part of the TRP pool is bound to albumin in the peripheral circulation [98,99]. As such, lowered albumin will decrease total TRP levels, thereby lowering brain TRP concentrations and maybe impacting serotonin synthesis in the brain [66,100,101]. In this respect, lowered albumin showed a negative correlation with depressive symptoms in patients with suicidal attempts [102]. Second, platelet uptake of TRP may be increased in MDD [103,104]. Third, increased free fatty acids, partially mediated by insulin, may affect TRP levels in MDD [48]. All the above-mentioned causes are probably responsible for reduced levels of TRP in severe affective disorders in the absence of upregulated TRYCAT pathway.

It should be stressed that patients with severe forms of affective disorders are probably treated with many antidepressants, mood stabilizers and antipsychotics, which may have substantial effects on IDO activity and TRYCATs levels [105,106]. Moreover, some antidepressants have anti-inflammatory properties by (a) inhibiting overactivated cell-mediated immunity and decreasing IFN-γ levels (a potent stimulator of IDO enzyme), therefore impeding stimulation of the TRYCAT pathway [107], and (b) reducing various APPs, such as haptoglobin, fibrinogen, C3, C4, and α-antitrypsin [108]. Furthermore, animal-based studies reported that valproate and citalopram negatively regulate the IDO and tryptophan 2,3-dioxygenase (TDO) enzymes and reserve TRP for the 5-hydroxytryptamine (5-HT) pathway [105,106]. Therefore, in severe affective disorders, many factors control the activity of the TRYCAT pathway.

### 7.3. Neurotoxic Indexes and TRYCATs

The third major finding of the current study is that the neurotoxicity index (including the composite score KYN+3HK+3HA+XA+QA+PA) is unaltered in patients with severe affective disorders. In addition, the KA/KYN ratio (another neurotoxicity index) was significantly decreased in patients with suicidal behavior and psychotic features but not in those with melancholia. The alteration in the KA/KYN ratio is probably due to decreased KA levels since we did not find any abnormality in KYN levels. We recently found that mild to moderate MDD/BD patients showed unchanged neurotoxicity composite scores and an increased KA/KYN ratio [40]. In line with the current study, previous studies showed a reduced KA/KYN ratio in suicidal depressed patients [81].

Nevertheless, severe affective disorder was accompanied by a significantly increased level of QA, a neurotoxic TRYCAT. Previous studies concerning QA were often inconsistent. For example, in the CSF of suicidal MDD patients, QA levels were elevated [41,42], while postmortem studies of suicide patients showed that in the CA1 and CA2/3 areas of the hippocampus, QA levels were either significantly decreased or unchanged [57]. In contrast, QA levels were increased in subgenual anterior cingulate cortex (sACC), and anterior midcingulate cortex (aMCC) areas of the brain of depressed patients with suicidal behaviors [79]. Moreover, in the plasma of suicidal MDD patients, QA levels were either significantly decreased [52] or unaltered [56]. No significant changes in QA levels were found in depressed patients with psychotic features or melancholia [54,60]. In the current study, we did not find any difference between peripheral and central QA levels or between patients with melancholia, psychotic features, and suicidal behavior, but, overall, there was a significant increase.

### 7.4. Neuroprotective TRYCATs

The fourth major finding of this study is that severe forms of affective disorders are linked to lower KA levels without a change in AA levels (after bias correction), even though the effect size was only computed in 3 studies. Recently, we found in mild to moderate MDD and BD patients that there was a peripheral reduction in KA and no changes in central levels [40]. In this regard, we did not find any significant difference among central and peripheral levels in the present study. Some previous papers indicated significantly lower central and peripheral KA levels [42,54]. However, other studies showed no aberration in KA levels, whether centrally or peripherally, in suicidal and melancholic MDD patients [41,60]. The main cause of lower KA is probably a decrease in KYN, the substrate of the KAT enzyme, although dietary factors cannot be excluded [109]. Recently, Steiner et al found a strong negative correlation between AA and severity of depression scores in unmedicated depressed females implying that the severity of depression is associated with lowered AA levels [110].

All in all, it appears that MDD/BD may be accompanied by lowered neuroprotection and consequent increased neurotoxicity. First, our findings indicating decreased KA but increased QA in severe affective patients indicate QA-based neurotoxicity [30]. Second, lowered levels of TRP (in itself an antioxidant) may lead to decreased antioxidant metabolites, namely serotonin, melatonin, 3HK and XA [111,112]. Moreover, serotonin enhances proper neuroplasticity and maintains healthy neurons [113,114]. Third, KA has a neuroprotective role by antagonizing the action of QA inhibiting excitatory receptors namely NMDA, α-amino-3-hydroxy-5-methyl-4-isoxazolepropionic acid (AMPA) and kainate glutamate ionotropic receptors, in addition, to impede alpha 7 nicotinic acetylcholine receptor (α7nAChr) and thereby decrease glutamate release [30,115]. Fourth, inhibition of the TRYCAT pathway results in decreased KYN, KA and XA levels, which is associated with an indirect increase in neurotoxicity since these metabolites exert an anti-inflammatory role by diminishing IFN-γ/IL-10 ratio [116]. In addition, KA, 3HK, 3HA, and XA display antioxidant properties [31,117].

## 8. Limitations

Some limitations should be noticed while interpreting the current findings. The lack of information concerning treatment histories restricted our potential to examine the impact of drugs on TRP, CAAs and TRYCATs levels. We could not examine the status of TRP and TRYCATs in the central nervous system because only a limited number of studies on CSF and brain were available. Furthermore, no studies assessed CAAs in MDD with psychotic features or suicidal behaviors, and, therefore, we could not examine CAAs levels and the TRP/CAAs ratio in these subgroups.

## 9. Conclusions

Figure 1 shows the summary of our findings. Severe affective disorders are accompanied by a decreased availability of TRP to the brain, whilst the TRYCAT pathway is not upregulated, probably due to the multiple treatments administered to those patients. However, there was a significant increase in neurotoxic QA and a significant decrease in neuroprotective KA, indicating increased neurotoxicity.

## Figures and Tables

**Figure 1 cells-11-03112-f001:**
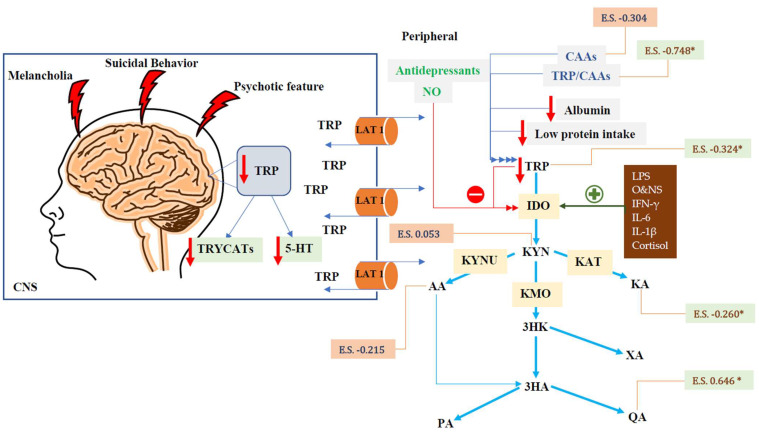
Summary of tryptophan catabolite (TRYCAT) pathway in severe affective disorders. *: significant difference. CNS: central nervous system. TRYCATs: tryptophan catabolites. E.S.: effect size. TRYCAT: tryptophan catabolite. LAT 1: large neutral amino acid transporter 1. IFN-γ: interferon-gamma. IL-6: interleukin 6. IL-1β: interleukin-1 beta. O&NS: oxidative and nitrosative stress. NO: nitric oxide. 5-HT: 5-Hydroxytryptamine. LPS: lipopolysaccharides. CNS: central nervous system. IDO: indoleamine 2,3 dioxygenase. TDO: tryptophan 2,3-dioxygenase. KAT: kynurenine aminotransferase. KMO: kynurenine 3-monooxygenase. KYNU: kynureninase. TRP: tryptophan. KYN: kynurenine. KA: kynurenic acid. 3HK: 3-Hydroxykynurenine. AA: anthranilic acid. XA: xanthurenic acid. 3HA: 3-hydroxyanthranilic acid. PA: picolinic acid. QA: quinolinic acid.

**Figure 2 cells-11-03112-f002:**
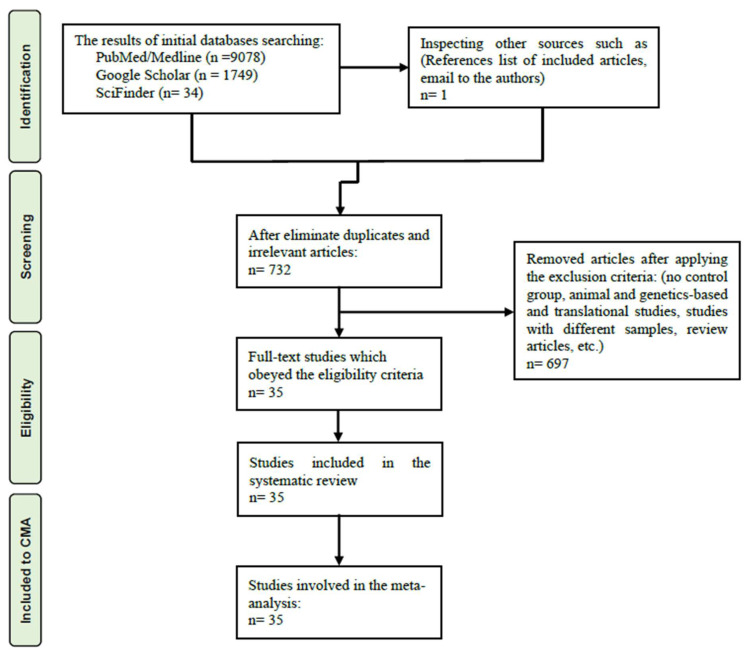
The PRISMA flow chart.

**Figure 3 cells-11-03112-f003:**
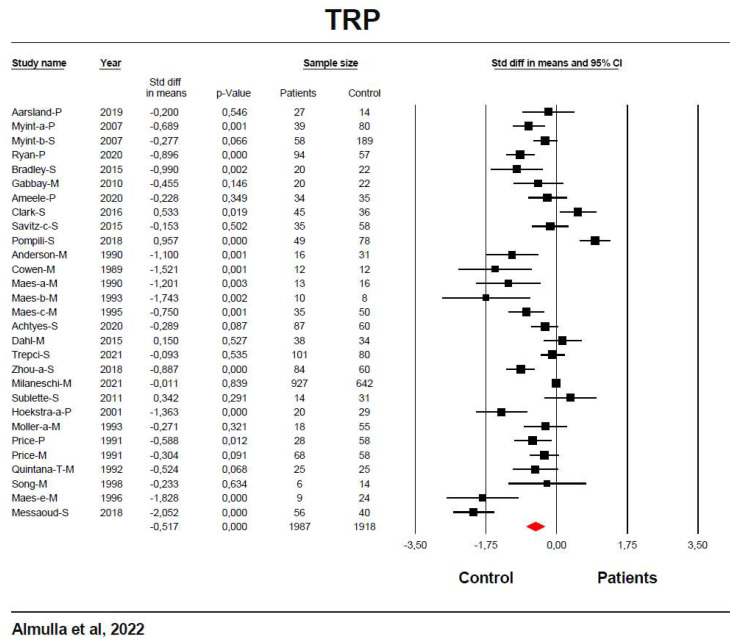
The forest plot of tryptophan (TRP) between severe affective disorder patients and healthy control.

**Figure 4 cells-11-03112-f004:**
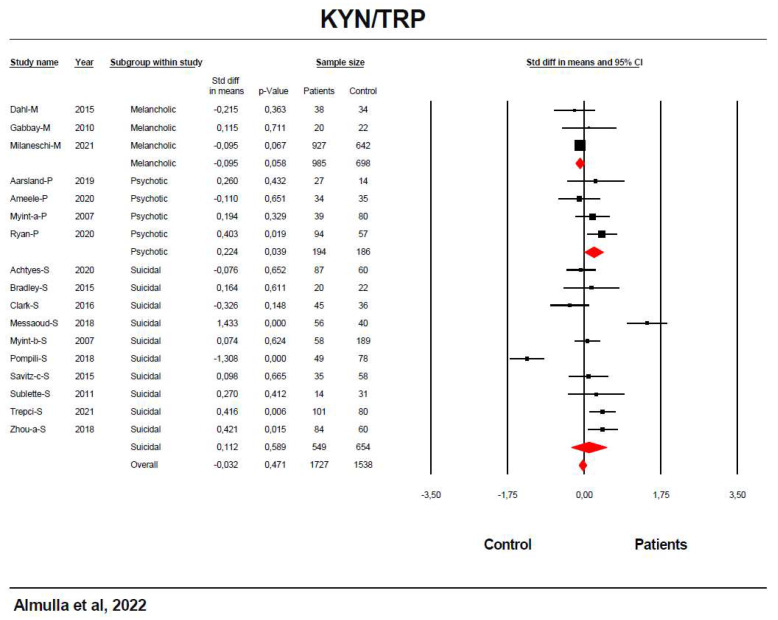
The forest plot of kynurenine (KYN)/tryptophan (TRP) ratio between severe affective disorder patients and healthy control.

**Table 1 cells-11-03112-t001:** The outcomes and number of patients with affective disorders and healthy control along with the side of standardized mean difference (SMD) and the 95% confidence intervals with respect to zero SMD.

Outcome Profiles	n Studies	Side of 95% Confidence Intervals				Patient Cases	Control Cases	Total Number of Participants
		<0	Overlap 0 and SMD < 0	Overlap 0 and SMD > 0	>0			
TRP	29	13	12	2	2	1987	1918	3905
TRP/CAAs	7	2	5	0	0	113	219	332
CAAs	5	2	2	1	0	86	163	222
KYN/TRP	17	1	5	7	4	1727	1538	3265
KYN	17	4	8	2	3	1727	1538	3265
KA/KYN	14	2	9	3	0	1619	1378	2997
(KYN + 3HK + 3HA + XA + QA + PA)	26	6	10	4	6	1874	1752	3626
KA	13	5	4	3	1	1563	1338	2901
AA	3	1	2	0	0	170	149	319
QA	14	2	2	6	4	1461	1131	2592

TRP: tryptophan. KYN: kynurenine. KA: kynurenic acid. 3HK: 3-Hydroxykynurenine. 3HA: 3-Hydroxyanthranilic acid. XA: xanthurenic acid. QA: quinolinic acid. PA: picolinic acid. AA: anthranilic acid. CAAs: competing amino acids (valine + phenylalanine + tyrosine + leucine + isoleucine).

**Table 2 cells-11-03112-t002:** Results of meta-analysis performed on several outcome (TRYCATs) variables with combined different media and separately.

Outcome Feature Sets	n	Groups	SMD	95% CI	z	*p*	Q	df	*p*	I^2^ (%)	τ^2^	Τ
TRP	29	Overall	−0.517	−0.735; −0.299	−4.650	<0.0001	226.434	28	<0.0001	87.634	0.282	0.531
TRP/CAAs ^#^	7	Melancholia	−0.617	−0.957; −0.277	−3.557	<0.0001	10.435	6	0.107	42.500	0.086	0.293
CAAs ^#^	5	Overall	−0.304	−0.674; 0.066	−1.612	0.107	6.676	4	0.154	40.086	0.070	0.264
KYN/TRP *	17	Overall	−0.032	−0.193; 0.003	−1.896	0.058	107.82	16	<0.0001	85.161	0.170	0.614
	3	Melancholia	−0.095	−0.193; 0.003	−1.896	0.058	0.714	2	0.700	0.000	0.000	0.000
	4	Psychotic	0.224	0.012; 0.436	2.068	0.039	3	3	0.392	0.000	0.000	0.000
	10	Suicidal	0.112	−0.119; 0.055	−0.720	0.471	93.734	9	<0.0001	90.398	0.377	0.614
KYN	17	Overall	−0.114	−0.352; 0.152	−0.935	0.350	126.513	16	<0.0001	87.353	0.203	0.450
KA/KYN *	14	Overall	−0.035	−0.117; 0.048	−0.824	0.410	26.181	13	0.016	50.346	0.028	0.167
	2	Melancholia	0.049	−0.050; 0.148	0.969	0.333	0.402	1	0.526	0.000	0.000	0.000
	5	Psychotic	−0.201	−0.416; 0.013	−1.838	0.066	0.912	4	0.923	0.000	0.000	0.000
	7	Suicidal	−0.231	−0.432; −0.030	−2.256	0.024	13.104	6	0.041	54.213	0.039	0.198
(KYN + 3HK + 3HA + XA + QA + PA)	26	Overall	0.048	−0.189; 0.284	0.396	0.692	209.842	25	<0.0001	89.086	0.295	0.543
KA	13	Overall	−0.260	−0.487; −0.034	−2.258	0.024	67.574	12	<0.0001	82.242	0.125	0.354
AA	3	Overall	−0.248	−0.485; −0.011	−2.055	0.040	2.115	2	0.347	5.432	0.003	0.051
QA	14	Overall	0.358	0.015; 0.701	2.044	0.041	134.272	13	<0.0001	90.318	0.343	0.585

* Significant difference between melancholia, psychotic feature and suicidal behavior. # The effect size was pooled from only melancholia. TRP: tryptophan. KYN: kynurenine. KA: kynurenic acid. 3HK: 3-Hydroxykynurenine. 3HA: 3-Hydroxyanthranilic acid. XA: xanthurenic acid. QA: quinolinic acid. PA: picolinic acid. AA: anthranilic acid. CAAs: competing amino acids (valine + phenylalanine + tyrosine + leucine + isoleucine).

**Table 3 cells-11-03112-t003:** Results on publication bias.

Outcome Feature Sets	Fail Safe n	Z Kendall’s τ	*p*	Egger’s *t* Test (df)	*p*	Missing Studies (Side)	After Adjusting	
							SMD	95%CI
TRP	−10.30	1.819	0.034	3.063 (27)	0.002	5 (Right)	−0.324	−0.548; −0.101
TRP/CAAs	−4.785	0.300	0.381	1.142 (5)	0.152	2 (Left)	−0.748	−1.069; -0.427
KYN/TRP (Psychotic)	1.827	0.339	0.367	0.764 (2)	0.262	-	-	-
KA/KYN (Suicidal)	−3.342	0.600	0.274	0.663 (5)	0.268	2 (Left)	−0.337	−0.541; −0.133
(KYN + 3HK + 3HA + XA + QA + PA)	−0.094	0.705	0.240	0.892 (24)	0.190	3 (Right)	0.210	−0.046; 0.467
KA	−5.072	0.549	0.291	0.692 (11)	0.251	-	-	-
AA	−2.199	0.000	0.500	0.422 (1)	0.372	1 (Right)	−0.215	−0.430; 0.00006
QA	3.846	0.985	0.162	1.618 (12)	0.065	3 (Right)	0.646	0.228; 1.065

TRP: tryptophan. KYN: kynurenine. KA: kynurenic acid. 3HK: 3-Hydroxykynurenine. 3HA: 3-Hydroxyanthranilic acid. XA: xanthurenic acid. QA: quinolinic acid. PA: picolinic acid. AA: anthranilic acid. CAAs: competing amino acids (valine + phenylalanine + tyrosine + leucine + isoleucine).

## Data Availability

The dataset generated during and/or analyzed during the current study will be available from the corresponding author (MM) upon reasonable request and once the dataset has been fully exploited by the authors.

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
