# Peer review of "The Tryptophan Catabolite or Kynurenine Pathway in a Major Depressive Episode with Melancholia, Psychotic Features and Suicidal Behaviors: A Systematic Review and Meta-Analysis"

_cells, 2022, doi:10.3390/cells11193112_

Round 1
Reviewer 1 Report
This meta-analysis is an important contribution to the field of kynurenines research.
Reviewer 2 Report
Comments on the review manuscript titled " The Tryptophan Catabolite or Kynurenine Pathway in a Major Depressive Episode with Melancholia, Psychotic Features and Suicidal Behaviors; a Systematic Review and Meta-Analysis".
Comments:
1. Some studies have focused on the kynurenine pathway in major depressive disorder and bipolar disorder for example, Marx et al., 2020 (PMID: 33230205). However, the novelty of the present study is that the authors have focused on the Major Depressive Episode with Melancholia, Psychotic Features, and Suicidal Behaviors. In the introduction, mention how this study differs and has novelty from previously published reviews and meta-analyses for the general audience.
2. Methodology and criteria of selection are well explained.
3. In figure-1 legend, the authors have just provided the abbreviation. Please provide some details of the pathway in the legend.
4. Overall, it is a very well-written systematic review and meta-analysis.
5. Check for minor grammar and spelling mistakes throughout the manuscript.
Reviewer 3 Report
Despite many meta-analyses related to depression and tryptophan derivatives, this publication brings together a new very interesting review on depression and its main symptoms melancholia, psychotic features, and suicidal behaviors. A very surprising and interesting conclusion, "Patients with affective disorders with melancholic and psychotic features and suicidal behaviors show normal IDO enzyme activity..." as one would have expected an increase in IDO activity in depression, however. The publication is a pleasure to read